# Can Bee Venom Be Used as Anticancer Agent in Modern Medicine?

**DOI:** 10.3390/cancers15143714

**Published:** 2023-07-21

**Authors:** Agata Małek, Maciej Strzemski, Joanna Kurzepa, Jacek Kurzepa

**Affiliations:** 1Department of Medical Chemistry, Medical University of Lublin, Chodźki 4a, 20-093 Lublin, Poland; jacek.kurzepa@umlub.pl; 2Department of Analytical Chemistry, Medical University of Lublin, Chodźki 4a, 20-093 Lublin, Poland; maciejstrzemski@umlub.pl; 31st Department of Radiology, Medical University of Lublin, Jaczewskiego 8, 20-090 Lublin, Poland; joannakurzepa@umlub.pl

**Keywords:** bee venom, melittin, cancer, tumor, apoptosis

## Abstract

**Simple Summary:**

The beneficial effects of natural substances that have been used in medicine for many years require both scientific confirmation and recognition of the mechanism of their action. In this paper, we present the current state of knowledge regarding the mechanisms of anticancer activity of bee venom in in vitro and animal model studies. So far, research shows strong anti-cancer potential of both crude bee venom and its main constituent, melittin, by inducing apoptosis and inhibiting the cell cycle without significantly affecting physiological cells. Increasingly frequent animal studies indicate the safety of venom doses that are effective in in vitro studies. This information can help plan future clinical trials.

**Abstract:**

Honey bee venom in its composition contains many biologically active peptides and enzymes that are effective in the fight against diseases of various etiologies. The history of the use of bee venom for medicinal purposes dates back thousands of years. There are many reports in the literature on the pharmacological properties of bee venom and/or its main components, e.g., anti-arthritic, anti-inflammatory, anti-microbial or neuroprotective properties. In addition, both crude venom and melittin exhibit cytotoxic activity against a wide range of tumor cells, with significant anti-metastatic activity in pre-clinical studies. Due to the constantly increasing incidence of cancer, the development of new therapeutic strategies in oncology is a particular challenge for modern medicine. A review paper discusses the various properties of bee venom with an emphasis on its anticancer properties. For this purpose, the PubMed database was searched, and publications related to “bee”, “venom”, “cancer” from the last 10 years were selected.

## 1. Introduction

Nature is an abundant source of many medications for both treating various diseases and alleviating their symptoms. Over the centuries, secondary metabolites from bacteria, fungi, insects or terrestrial plants have played an essential role in traditional medicine, but bee products have greatly distinguished themselves in chemical biodiversity, providing not only an efficient, but also a widely available and renewable source of therapeutic substances. The honey bee (*Apis mellifera* L.) is a key pollinator species in both natural and agricultural environments [1]. Various types of products produced by bees have been known for their health-promoting properties since ancient times. Natural bee products used in the treatment and prevention of various diseases include honey, bee pollen, propolis, royal jelly, bee pollen, beeswax, as well as bee venom (BV) also known as apitoxin [2]. In traditional Chinese medicine from approx. 1000–3000 BC, bee venom has been used therapeutically, initially mainly for arthritis or other joint and muscle ailments [3]. BV is a complex mixture of several compounds with a number of pharmacological properties. In high concentrations, BV can trigger inflammation, pain and allergic reactions. However, based on studies conducted on animal models, it can be concluded that low concentrations of BV (up to 5 μg/mL) can produce beneficial therapeutic effects such as anti-inflammatory, antinociceptive, anticancer or antimicrobial activity [4]. Apitoxin also has a neuroprotective effect. In mouse models, the effect of BV on reducing progression and improving cognitive functions in the course of such neurodegenerative diseases as Parkinson’s disease, Alzheimer’s disease or amyotrophic lateral sclerosis has been demonstrated [5]. The ability to neutralize free radicals by BV also includes it in the catalog of substances with antioxidant properties, having potential use in the therapy of physiological and pathological conditions combined with increased oxidative stress [6]. The main topic discussed in the article is the anti-cancer effect of BV. This is an important issue because more and more in vitro and animal studies confirm the inhibitory effect of BV and its main component, melittin, on the development of various types of cancer giving reason for further research in this direction. The versatile properties of BV make it an attractive substance that could be used in various branches of medicine.

### 1.1. What Are the Components of Bee Venom?

BV is a secretion from the venom gland of worker bees that is supposed to have a defensive function against predators [5]. It is an odorless and colorless liquid with an acidic pH of 4.5–5.5. One drop of bee venom consists of 88% water and only 0.1 μg of dry matter [7]. Among the various types of products of natural origin, BV deserves special attention due to the complexity of its chemical composition and the potential inherent in its biological activity. BV is a rich source of secondary metabolites, such as: peptides, including melittin, apamin, mast cell degranulating peptide (MCD) or adolapin, enzymes such as phospholipase A_2_ (PLA_2_) and hyaluronidase, as well as amino acids and volatile compounds [5]. The main components of BV are melittin, which constitutes about 50% of the dry weight of the venom, and PLA_2_, whose content is about 12% (Figure 1) [8]. Of the organic acids, citric acid has the highest concentration in apitoxin, while glutaric acid and kynurenic acid are the least abundant organic acids present in honey BV [9]. BV is characterized by composition variability depending on many factors, such as age, strain, social position of bees or geographical location and season [10].

Melittin is a water-soluble, amphipathic linear polypeptide consisting of 26 amino acids, with a mass of 2840 Da and a chemical formula of C_131_H_229_N_39_O_31_. Its N-terminal region is hydrophobic, while the C-terminal region is hydrophilic [12]. Melittin affects cell membranes and reduces their surface tension, stimulates smooth muscles, increases capillary permeability, reduces blood coagulability; in higher doses it has a pro-inflammatory and hemolytic effect [13]. It seems that melittin is the main component of the BV responsible for the cytotoxic effect on cancer cells, which is why many studies are conducted using this peptide rather than crude venom.

BV phospholipase A_2_ is a single polypeptide chain of 128 amino acids, containing four disulfide bridges. PLA_2_ is a hydrolytic enzyme capable of specifically cleaving ester bonds of phospholipids at the sn-2 position. There is a synergistic interaction between bee venom PLA_2_ and melittin. The effect of PLA_2_ can be enhanced by melittin, which has been demonstrated during the process of erythrocyte lysis. This phenomenon is based on the participation of melittin in the exposure of membrane phospholipids to the catalytic site of the enzyme by opening melittin-induced channels [5,14].

Despite BV is a toxin, its application to humans is a relatively safe therapeutic strategy in the absence of allergy to it. The lethal dose for an adult is estimated at about 2.8 mg of venom per kg of body weight. One bee has only 0.15–0.30 mg of venom. Therefore, the number of stings necessary to cause death is as high as about 1300 [15]. Administering particular components of BV, such as melittin, sometimes can be more effective and safer than treating with the crude BV. Various chemical modifications of BV in order to remove potentially harmful compounds could potentially reduce its toxicity [16,17]. The toxicity of melittin can be reduced, for example, by conjugation with polyethylene glycol (PEGylation) and shortening and synthesis of the peptide using dextrorotatory D-amino acids. By minimizing the toxicity of BV, it becomes possible to use higher concentrations of apitoxin, and thus achieve a better therapeutic effect [4]. 

### 1.2. How Does the Body React to Bee Venom?

BV can cause both mild and severe clinical effects depending mainly on the number of stings. Clinical symptoms after a sting can be divided into local inflammatory reactions, allergic reactions, anaphylactic shock and systemic toxicity (Figure 2) [18]. People with atopy and a family history of BV allergy have a higher incidence of severe sting reactions [19]. PLA_2_ is considered the main compound that induces sensitization of mast cells to IgE antibodies, although hyaluronidase and melittin are also responsible for this effect [18,20]. Local inflammatory reactions are characterized by pain, swelling, erythema and itching at the sting site. These reactions occur in most non-allergic individuals and usually subside within 24 h [21]. Allergic reactions to BV are IgE-dependent and are classified as type I hypersensitivity reactions. They appear about 10 min later after the sting, and the symptoms may vary in severity [20]. People with allergies may develop hives, pruritus, angioedema, vomiting and diarrhea [22]. In some cases, allergic reactions can lead to an anaphylactic reaction causing bronchospasm and anaphylactic shock leading even to death [21]. Systemic toxicity occurs independently of immune mechanisms, while being dependent on the volume of venom introduced into the body. Systemic toxic reactions are always considered severe and are triggered by multiple bee stings (approximately 50 simultaneous stings). Patients experiencing a systemic toxic reaction experience dizziness, nausea, vomiting or diarrhea. In such cases, myocardial damage, liver damage, and rhabdomyolysis leading to acute renal failure and coma may occur [18,23]. The broad therapeutic potential of BV should not cause one to underestimate possible side effects and the possibility of an allergic reaction, as evidenced by the case of a Spanish woman who died due to an anaphylactic reaction to acupuncture from live bees described in recent years [24]. Unfortunately, there is no antivenom that can be used for people who are stung and are allergic to BV. This is due to the low immunogenicity of BV proteins (e.g., melittin), which makes it difficult to immunize animals in order to obtain high antibody titers in their plasma and, consequently, significantly complicates the work on bee antitoxin [25]. 

## 2. Can Bee Venom Cure Cancer?

Already in the early 1950s, the effect of apitoxin on the colchicine-induced carcinogenesis process was described [26]. In turn, the results of the analysis of causes of death for professional beekeepers compared to the rest of the population suggested the oncoprotective potential of BV, especially in the case of lung cancer [27]. From the early 1980s, publications containing the results of research on the anti-cancer properties of BV began to appear. Currently, the scientific literature contains many publications on preclinical studies concerning the effect of venom and its main components, most often melittin, on the growth of cancer cells. In order to present the latest reports on this topic, we have collected original publications from the last ten years, present in the PubMed database. The publication selection flowchart is shown in Figure 3. Out of 141 publications from 2013–2023 searched for by the words “bee”, “venom”, “cancer” present in the title or abstract, 102 were selected, rejecting review publications. From the rest, 57 studies on the impact of venom on cancer development were selected rejecting publications that were not directly related to the study of the anticancer properties of BV. Fifty-one publications concerned studies using cell cultures of various cancer cell lines, 5 studies were conducted on animals, and one study was performed on both cultures and an animal model. Basic information on studies is presented in Table 1 and in Table 2. The number of studies on the potential use of BV in the treatment of cancer has increased significantly in recent times. By the end of May 2023, eight publications from this year have already been published in the PubMed database, the most among the analyzed years (2013–2023), despite the fact that it is not yet halfway through the year. This indirectly shows the interest of the world of science in new agents with a possible anti-cancer effect, including BV.

Among the studies on cell cultures, only one did not show a statistically significant cytotoxic effect on cancer cells (human glioblastoma) [32]. However, the study indicates an influence of BV on the decrease in the pro-inflammatory cytokines level. Other investigations reveal varying degrees of cytotoxic effects of both crude venom and melittin on different tumor cell lines. Most of the authors analyze the possible molecular mechanisms of BV action, but some of them are limited only to the statement of the toxic effect of the venom on cancer cells. Both crude BV and melittin have been found to have a pronounced cytotoxic effect in breast cancer in vitro studies. The authors also point to the synergy of BV with cisplatin [57], hesperidin and piperine [60], 5-fluorouracil and fluphenazine [37]. BV or melittin, alone or in combination with above mentioned chemotherapeutic agents, induced apoptosis in breast cancer cells, showed antiproliferative activity, and inhibited cell migration [34,66,74]. Tetikoğlu et al. [28] indicate the genotoxicity of BV to both breast cancer cells and physiological cells by the interaction of phosphorylated H2A histone family member (XγH2AX) and β-actin. Typically, BV caused an imbalance between pro-apoptotic and anti-apoptotic factors. A common observation in various cell types was the upregulation of Bax, p53, p21 and the downregulation of Bcl2, cyclin A and cyclin B [51]. Also, an increase in the concentration of cleaved caspase-3, the main enforcer of apoptosis, was observed when cells were incubated with BV [68]. BV and melittin affect many intracellular pathways, which inhibition also induces apoptosis. An example is the inhibition of the PI3K/Akt/mTOR pathway by melittin in breast cancer cells [74] or in liver cancer cells [73]. The inhibition of PI3K/Akt/mTOR and MAPK pathways under the influence of BV and melittin in malignant melanoma cells is indicated by the study by Lim et al. [56]. In two studies on ovarian cancer cells, Alonezi et al. point to yet another mechanism of melittin action under the influence of which there is a reduction in the levels of metabolites in tricarboxylic acid cycle, oxidative phosphorylation, purine and pyrimidine metabolism, and the arginine/proline pathway. In addition, the authors noticed the decreased levels of carnitines, polyamines, adenosine triphosphate (ATP) and nicotinamide adenine dinucleotide (NAD^+^) under the influence of melittin [63,65]. Erkoc et al. [34] reports the induction of calcium signaling apoptosis and inhibition of cAMP under the influence of melittin in breast cancer cells which leads to apoptosis and antiproliferative effect. BV also affects cell morphology leading to DNA and protein fragmentation [58]. Research by Li et al. [35], conducted on lung cancer cells, indicate that melittin increases the production of reactive oxygen species (ROS) and the accumulation of intracellular iron, while disrupting the functioning of glutathione peroxidase 4 (GPX4), which leads to mitochondrial damage and apoptosis called ferroptosis. Some studies in recent years concern the influence of BV on the migration of cancer cells. Growth of the tumor mass is a complicated process in which the stromal tissue is gradually destroyed by proteolytic enzymes. Of these, matrix metalloproteinases (MMPs) play a major role in this process. BV reduces the activity of MMP-2 and MMP-9 in glioblastoma cells in a dose-dependent manner, but not in healthy hippocampal cells [38]. It is worth mentioning that BV active ingredients, unlike many other therapeutic molecules, have the ability to penetrate the blood-brain barrier and therefore can be considered as a potential agent for the treatment of CNS diseases in the future [84]. Also, studies on gastric cancer cells indicate that melittin inhibit MMP-2 and MMP-9 activity [43]. A similar effect is exerted by BV and melittin in hepatocarcinoma cells [42]. Reducing tumor invasion is also associated with inhibition of angiogenesis. Shin et al. [78] showed that melittin inhibits hypoxia-inducible factor 1-alpha (HIF-1ɑ), a transcription factor responsible for the expression of vascular endothelial growth factor (VEGF), a strong stimulator of blood vessel formation. Similar results were obtained by Zhang et al. [62] in non-small cell lung cancer.

The main idea of the in vitro studies evaluating the effect of BV on cancer cells is to compare the obtained results with the effect on physiological cells. The demonstration of the selective effect of the venom on cancer cells should be a necessary condition to continue the research using other models. However, two strategies for evaluating the effectiveness of BV are evident in research studies. One of them compared the effect of BV on cancer cells with the effect exerted on physiological cells. The second strategy compares the effect of BV on tumor cells with the effect exerted by an agent with known cytostatic properties like cisplatin for example. The cytotoxic activity of BV against tumor cells varies depending on the type of cell line and the time of cell incubation with BV. In our previous studies, in which glioblastoma cell lines were used, the cytotoxic effect of crude BV on the 8-MG-BA cell line, after 24 h of incubation, started at a concentration of 0.5 µg/mL. Per GAMG cell line, the inhibitory effect started from a concentration of 1.0 µg/mL. At the same time, the cytotoxic effect of BV on physiological cells, HT-22 hippocampal cells, was revealed at the BV concentration of 1.25 µg/mL. However, the venom much more rapidly decreased the number of viable pathological cells; a concentration of 1.25 µg/mL reduced the number of 8-MG-BA cells by more than 80% but the number of physiological HT-22 cells decreased about only 15% at this concentration. For longer incubation (48 h or 72 h), the difference between BV cytotoxic activity against tumor cells vs. physiological cells was even more pronounced at higher concentrations [38]. In other study conducted on the U87 human glioblastoma line, effective cytotoxic effects started at a concentration of 5 µg/mL. This study did not take into account the effect of venom on physiological cells [24]. Gajski et al. revealed the cytotoxic effect of BV on glioblastoma A1235 cells at a concentration of 20 µg/mL [67]. However, 2 µg/mL of the crude BV was able to induce a significant decrease in cell viability (above 60%) in human colon cancer HCT116 cells [45]. Another study using the same colon cancer cell line demonstrated the cytotoxic effect of BV at a concentration of 1 µg/mL. In addition, BV did not have any toxic effect on FHC colon epithelial normal cells even at a concentration of 10 µg/mL [68].

The observed concentrations of BV exerting an inhibitory effect on cancer cells could be too low to consider the usefulness of systemic treatment without significant side effects. The high ability to cause hemolysis by melittin and its rapid metabolism could be a potential barrier to using BV. The cytotoxic effect of BV on tumor cells in in vitro studies is revealed at a concentration of >1 µg/mL in the medium (depending on the type of cell line). Theoretically, to obtain such a concentration of BV in the body fluids of a 75 kg human, approximately 75 mg of venom would have to be administrated. However, in animal studies, a venom dose from 0.5 mg/kg bw to 1.0 mg/kg bw was successfully used [81,83]. Other inconveniences of systemic treatment could be avoided by using directly intratumoral administration of BV. This method has a low toxicity and a high therapeutic index [79,85]. Also, the loading of BV on polymers or nanoparticles decreases the side effects of systemic treatment (this issue is widely discussed in ref. [86]).

Studies on cell cultures allow the assessment of the molecular mechanisms of a drug’s action, but they do not provide an answer to the behavior of the cancer in the natural environment of the host. Therefore, animal models allow both the assessment of tumor growth in the body’s conditions as well as the ability to form distant metastases. There have been far fewer animal studies than in vitro studies on anticancer activity of BV. It has been shown that BV reduces MMP-2 and MMP-9, decreases VEGF, tumor necrosis factor (TNF) and nitric oxide (NO) levels leading to suppressing tumor proliferation and inhibiting angiogenesis in the Ehrlich ascites carcinoma mouse model [81]. The reduction of VEGF levels in lung cancer in mice under the influence of melittin was also reported by Lee et al. [82]. Inhibition of colorectal cancer metastases growth in the mice model under the influence of melittin was observed by Rocha et al. [79]. Restoration of histological changes in ovarian and breast cancer rat models together with decreasing of serum MMP-1, reduction of NF-κB, and TNF was described by El-Beltagy et al. [80]. 

Both in vitro and animal studies indicate the anti-cancer effect of BV and melittin. The observations made so far bring us closer to starting clinical trials on the use of BV in oncology in the future. However, there is no registered clinical trial (ongoing or completed) dedicated to evaluating BV as a clinically useful anti-cancer substance on the www.clinicaltrials.gov (accessed on 1 June 2023) web page. Perhaps further evidence of the positive effects of BV will contribute to the start of research on humans. As mentioned above, the impact of BV on the body is multidirectional, and the effect of its individual components on the entire human body requires detailed research. It is not known whether BV is able to reach therapeutic concentrations in tumor tissue without causing negative symptoms. It should be remembered that in addition to the anti-cancer effect, the venom has other systemic activities which may also contribute to the anticancer activity of BV.

### Anti-Inflammatory, Antioxidative and Antimicrobial Effects of Bee Venom

BV PLA_2_ may exhibit anti-inflammatory properties at low concentrations, however it exerts pro-inflammatory potential at higher concentrations. BV also contains substances that are typically pro-inflammatory, such as MCD-peptide and histamine [16,17]. On the other side, melittin possesses the ability to inhibit pro-inflammatory cytokines such as CXCL-8, interleukin (IL-6), TNF or interferon-γ (IFN-γ). It has been shown that melittin downregulates signaling pathways involved in activating inflammatory cytokines, including nuclear factor kappa B (NF-κB), protein kinase Akt and extracellular signal-regulated kinase (ERK1/2) in *Porphyromonas gingivalis* lipopolysaccharide-treated human keratinocytes. By blocking major signaling pathways of pro-inflammatory cytokine induction, melittin inhibits the activity of these molecules, resulting in decreased inflammation within different organs [87]. Studies conducted in a mouse model suggest that melittin treatment improves the anti-inflammatory capacity of the central nervous system (CNS) proteasome in a mouse model of amyotrophic lateral sclerosis (ALS) [88,89]. Crude BV can also protect from liver fibrosis, which is the result of chronic organ damage through an anti-inflammatory and anti-apoptotic mechanism, as demonstrated both in vivo and in vitro studies [90]. The anti-fibrotic effect of BV in a mouse model is based on a significant reduction in the levels of alanine aminotransferase (ALT) and aspartate aminotransferase (AST), as well as inhibition of the expression of pro-inflammatory cytokines [91]. BV can also reduce the severity of symptoms in atopic dermatitis, the most frequent allergic chronic inflammatory skin disease. A potential mechanism of BV’s anti-atopic effect is the inactivation of the complement system through the induction of CD55 [92]. Finally, BV acupuncture has been shown to relieve pain, through its influence on cytokine concentrations. Rats subjected to experimental spinal cord injury and then treated with BV acupuncture showed a decrease in serum concentrations of such pro-inflammatory cytokines as IL-6 and IL-1β, while concentrations of anti-inflammatory cytokines: IL-4 and IL-10 increased [93].

The reports indicating that BV exerts an effect on oxidative status are available in the literature. However, the obtained results are not compatible and sometimes indicate the opposite effect, antioxidative or pro-oxidative properties of BV. Interestingly, previous studies have shown that the antioxidant properties are not directly related to any of the individual components of the BV, and the strongest properties are presented by crude venom [94]. BV assessed by the 1,1-diphenyl-2-picrylhydrazyl (DPPH) assay method revealed that its antioxidant activity (depending on the concentration used) is comparable to the activity of vitamin C [95]. The mixture prepared with propolis, royal jelly and BV significantly increased the antioxidative enzymes activity (superoxide dismuthase, SOD; glutathione peroxidase, GPx; and catalase, CAT) with simultaneously decrease in malondialdehyde (MDA, biomarker of unsaturated fatty acids oxidation) level in the dexamethasone model of hypertensive rats [96]. However, this study used the mixture of bee-derived compounds, not only BV, and obtained results could come from other constituents of the mixture. One of the factors that increase the body’s oxidative stress is ionizing radiation, which is also used in cancer treatment or radiological diagnostics. Photon radiation (gamma and X-rays) has a negative effect on organisms mainly by inducing radiolysis of water molecules, and the ROS formed in this process initiates cascades of free radical reactions [97]. El Adham et al. showed that BV administered at a dose of 0.05 mg/kg bw significantly reduced the concentration of 8-oxo-2’-deoxyguanosine (8-oxo-dG, an oxidized derivative of deoxyguanosine, one of the major products of DNA oxidation), increased the activity of SOD, GPx, CAT and reduced glutathione (GSH) in rats subjected to 6 Gy gamma irradiation—indicating the antiradiative properties of BV [98]. Another study showed the melittin enhanced the antioxidant defense pathway by regulating the nuclear translocation of nuclear factor erythroid 2-like 2 (Nrf2) thus upregulating the production of the heme oxygenase-1 (HO-1), an important cellular antioxidant enzyme [99]. The ability to scavenge free radicals demonstrated during in vitro studies does not yet indicate that the substance in in vivo conditions, especially the substance with pro-inflammatory properties, is an effective agent reducing oxygen stress. Therefore, the antioxidant properties of BV are not fully explained, taking into account that the BV induces inflammatory processes in high concentrations and indirectly intensifies oxidative stress, and on the other hand, having antioxidant compounds can limit this process.

The last of the discussed properties of BV is its antimicrobial activity. BV is an example of a substance that exhibits significant antimicrobial activity against bacteria, viruses and fungi both in vitro and in vivo [100]. BV (but also melittin and PLA_2_) has the ability to inhibit the multiplication of bacteria belonging to different genera due to affecting the bacterial cell membranes. The formation of pores leads to the loss of the integrity of membranes leading to cell lysis [5]. It has been shown that melittin more easily penetrates the peptidoglycan layer of Gram-positive bacteria compared to Gram-negative cells, which have an additional layer of lipopolysaccharide. The presence of a proline residue at position 14 plays a key role in the antimicrobial activity of melittin. It has been shown that the lack of this amino acid significantly reduces the antimicrobial activity of the compound [101]. Melittin exhibits broad antimicrobial activity against a wide variety of bacteria, including both methicillin-sensitive *Staphylococcus aureus* (MSSA) and methicillin-resistant *Staphylococcus aureus* (MRSA), a common cause of nosocomial infections such as methicillin-resistant *Staphylococcus aureus* and *Enterococcus* spp. [102]. In vitro studies have shown that BV and melittin exert antibacterial properties against all morphological forms of the pathogen *Borrelia burgdorferi*, bacteria causing Lyme disease. In addition, apitoxin was also effective against bacterial biofilms of *B. burgdorferi* that are resistant to antibiotic therapy. The reports demonstrate the potential use of BV and its components against *B. burgdorferi*, offering hope for an effective treatment specific to relapses [103,104]. In the case of combining treatment with antibiotics (vancomycin, oxacillin and amikacin) or chemotherapeutics with the simultaneous use of BV, the phenomenon of synergistic therapeutic effects is observed. The synergistic effect contributes to the reduction of the doses of therapeutic agents used, and thus to the reduction of their side effects, as well as the reduction of drug resistance that has recently been growing in bacteria [100]. Moreover, BV turned out to be an interesting source of antiviral peptides [105]. Co-incubation of non-cytotoxic concentrations of BV or melittin was shown to significantly inhibit the replication of influenza A virus (PR8) and respiratory syncytial virus (RSV) [106]. Melittin showed antiviral activity against HIV-1 infected T-cell lymphoma cells. The cell line treated with melittin showed an almost complete absence of viral particles [107]. Melittin was also tested against herpes simplex virus: HSV-1 and HSV-2, showing significant inhibition of viral replication at relatively low concentrations of the compound. It is believed that the mechanism of action of peptides against enveloped viruses is based on the lysis of their envelope [108,109]. In addition, BV and its components are effective against non-enveloped viruses (e.g., enterovirus 71 or Coxsackie virus). The basis of apitoxin activity against these types of viruses is not precisely characterized, although it is assumed that it is based on the effect directly on the surface of the virus. It is worth adding that bee venom and its components can stimulate the synthesis of type I interferon, and thus inhibit viral replication in the host cell [106].

It was also noticed that in Wuhan, China, beekeepers, as a group exposed to frequent bee stings, are more resistant to COVID-19 infection, probably due to the regulation of the immune response and the increase in the titer of IgE and IgG antibodies [110]. It has been shown that BV regulates the immune response by stimulating the differentiation of cells expressing the Foxp3 (forkhead box P3) transcription factor, which is crucial for the development of regulatory T cells and their acquisition of suppressor activity. This effect has been observed in human CD4+ regulatory T cells and mature CD4+ thymocytes. These regulatory T cells play an important role in the immune response against SARS-CoV (severe acute respiratory syndrome coronavirus) infection [111,112]. However, only future research can verify whether BV and its components can be used as an activator of the immune system in the course of this infection.

## 3. Conclusions

Honey BV in its composition contains many biologically active peptides and enzymes that are effective in the fight against diseases of various etiologies. The history of the use of apitoxin for medicinal purposes dates back thousands of years. There are many reports in the literature on the pharmacological properties of bee venom and/or its main components, e.g., anti-arthritic, anti-cancer, anti-inflammatory, anti-microbial or neuroprotective properties. Both BV and melittin exhibit cytotoxic activity against a wide range of tumor cells, with significant anti-metastatic activity. The number of studies on the impact of BV on various types of cancer is increasing. The mechanisms of action of BV and its components are getting better and better known. However, there are no clinical trials on humans that could confirm the clinical effectiveness of this natural substance and assess the safety of its administration. Currently, the possibilities of clinical applications of BV are still distant, but the ongoing research on this topic undoubtedly brings closer the recognition of BV and its components as promising candidates for use in therapeutic strategies. Due to the multidirectional action of apitoxin, further research should focus on the cellular and molecular mechanisms of action of BV.

## Figures and Tables

**Figure 1 cancers-15-03714-f001:**
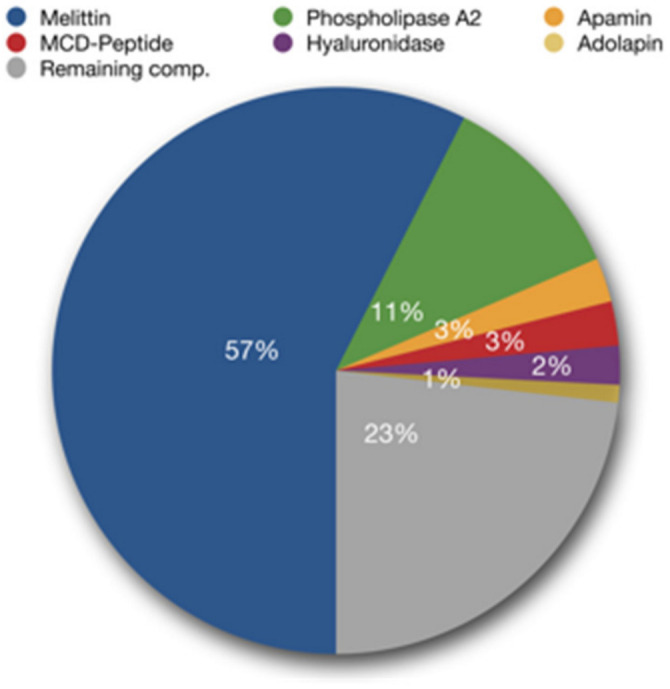
Main bee venom components (adapted from Bee Venom Lab LLC, Georgia). The composition of the venom depends on many factors, including the region of the world and the time of year when the venom is collected [11].

**Figure 2 cancers-15-03714-f002:**
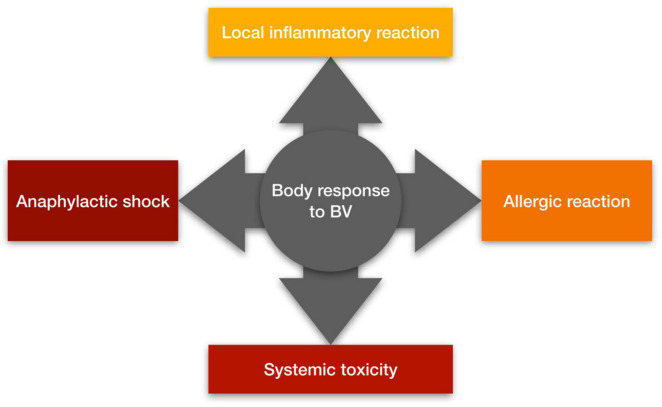
The body’s response to bee venom (prepared on the basis of [18]).

**Figure 3 cancers-15-03714-f003:**
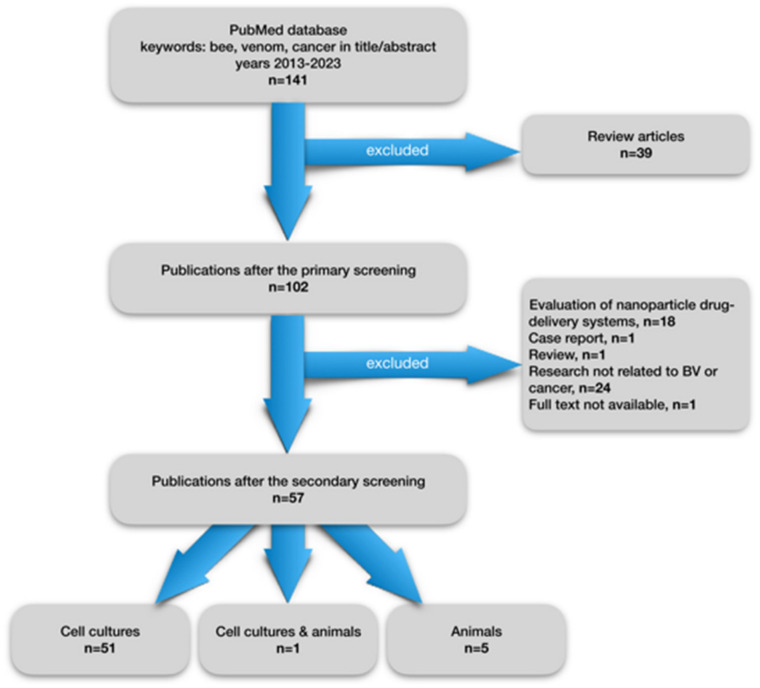
Flowchart showing a selection of articles from the last 10 years on preclinical studies on the effect of bee venom on tumor development.

**Table 1 cancers-15-03714-t001:** Original in vitro research on bee venom, melittin or PLA_2_ impact on cancer cells published in 2013–2023, present in the PubMed database. BV—Bee Venom, cAMP—cyclic Adenosine MonoPhosphate, CAT—catalase, DR—death receptor, EGF—endothelial growth factor, FAK—focal adhesion kinase, GPx4—glutathione peroxidase 4, HIF-1ɑ—hypoxia-inducible factor 1ɑ, HPV—human papilloma virus, IFN-γ—interferon gamma, IL-10—interleukin 10, JNK—c-Jun N-terminal kinases, PARP—poly ADP ribose polymerase, PLA_2_—phospholipase A_2_, PTEN—phosphatase and tensin homolog, MAPK—mitogen-activated protein kinase, MDA—malonyldialdehide, MMP—matrix metalloproteinase, mTOR—mammalian target of rapamycin, NF-κB—nuclear factor kappa B, NO—nitric oxide, SOD—superoxide dismuthase, TCA—tricarboxylic acid, TNBC—triple negative breast cancer, TNF—tumor necrosis factor, TRPM2—transient receptor potential cation channel subfamily M2, VEGF—vascular endothelial growth factor.

No.	Author	Year	Cell Culture	Species	BV Component or Crude BV	Molecular Mechanism of Acting	Effect
1	Tetikoğlu [28]	2023	Breast cancer	Human	BV	Interaction of γH2AX and β-actin	Genotoxicity
2	Obeidat [11]	2023	Leukemia	Human	BV, melittin	Modulation NF-κB and MAPK pathway, CDK4 inhibition	Apoptosis, necrosis, cell cycle arresting
3	Sevin [29]	2023	TNBC	Human	BV	Unknown	Apoptosis
4	Hwang [30]	2023	Lung cancer	Human	BV	Modulation of expression PARP, caspase-9, p53, Bcl2, Box	Cell death, cell cycle arresting
5	Yu [31]	2023	Lung cancer Glioblastoma TNBC Liver cancer	Human	BV	Inhibition of mTOR pathway	Autophagy induction
6	Sevin [32]	2023	Glioblastoma	Human	BV	Decrease of pro-inflammatory cytokine levels	Lack of cytotoxic effect
7	Ertilav [33]	2023	Glioblastoma	Human	Melittin	Stimulation of TRPM2 channel, prooxidative effect	Apoptosis
8	Erkoc [34]	2022	TNBC	Human	Melittin	Induction of calcium signaling apoptosis, inhibition of cAMP	Apoptosis, anti-proliferative effect
9	Li [35]	2022	Lung cancer	Human	Melittin	Upregulation of ROS production, increasing of intracellular ferrum level, disruption of GPx4, mitochondria damage	Apoptosis (ferroptosis)
10	Zhao [36]	2022	Pancreatic cancer	Human	BV	Modulation of cyclins and cyclin-dependent kinases (CDKs) expression, p53-p21 pathway activation	Apoptosis, cell cycle arresting
11	Duarte [37]	2022	Colon cancer Breast cancer	Human	BV	Unknown	Cytotoxic together with 5-FU and fluphenazine
12	Małek [38]	2022	Glioblastoma	Human	BV	Reduction of MMP2 and MMP9 secretion	Cytotoxic
13	Yaacoub [39]	2022	Cervical cancer	Human	BV, melittin	Unknown	Cytotoxic
14	Lischer [40]	2021	Breast cancer	Human	Melittin	Unknown	Cytotoxic
15	Gasanoff [41]	2021	Leukemia	Human	Melittin	Melittin-induced decline of mitochondrial bioenergetics	Cytotoxic
16	Mansour [42]	2021	Hepatocellular carcinoma	Human	BV, melittin	Upregulation of p53, Bax, Cas3, Cas7, PTEN. Downregulation Bcl-2, Cyclin-D1, Rac1, Nf-κB, HIF-1a, VEGF, MMP9. Oxidative stress induction.	Cell cycle arresting, apoptosis
17	Huang [43]	2021	Gastric cancer	Human	Melittin	MMP2 and MMP9 activity inhibition, decreasing of Wnt/BMP and MMP-2 signaling pathway activity. Inhibition of adhesion molecules.	Cytotoxic, adhesion and invasion inhibition
18	Lebel [44]	2021	Glioblastoma	Human	BV, melittin	Influence on Bak, Bax and Cas3	Apoptosis, necrosis
19	Yaacoub [45]	2021	Colon cancer	Human	BV, melittin, PLA2	Unknown	Synergistic activity of melittin and PLA2, cytotoxic
20	Borojeni [46]	2020	Cervical cancer Breast cancer	Human	BV	Unknown	Apoptosis
21	Kreinest [47]	2020	Hodgkin Lymphoma	Human	Melittin	Unknown	Cytotoxic, increase sensitivity of cisplatin
22	Grawish [48]	2020	Head and neck squamous cell carcinoma	Human	BV	Upregulation of Bax, downregulation of Bcl2 and EGFR, influence on cell cycle	Cytotoxic, cell cycle arresting, increasing of cisplatin activity
23	Sangboonruang [49]	2020	Malignant melanoma	Human	Melittin	Upregulation of cytochrome c and its translocation of cytosol, up regulation of Cas3 and Cas9. Reduction of EGFR expression.	Apoptosis
24	Salama [50]	2020	Liver carcinoma Breast cancer Cervical cancer	Human	BV	Influence on IL-10, TNF, IFN-γ. Elevation of Cas3 level.	Apoptosis
25	Kim [51]	2020	Cervical cancer	Human	BV	Increase in p53, p21, p27, Bax. Decrease in cyclin A, cyclin B, Bcl-2, Bcl-XL. Influence on caspases and intercellular signaling pathways.	Cytotoxic to HPVpositive cervical-cancer cell lines
26	Ceremuga [52]	2020	Leukemia	Human	BV	Effect on mitochondrial membrane potential, Annexin V binding and Caspases 3/7 activity	Apoptosis
27	Jeong [53]	2019	Non-small cell lung cancer	Human	BV	Inhibition of EGF-induced F-actin reorganization and cell invasion, inhibited EGF-induced ERK, JNK, FAK and mTOR phosphorylation	Cytotoxic
28	Soliman [54]	2019	Gastric cancer Colon cancer	Human	Melittin	Membrane affecting	Cytotoxic
29	Shaw [55]	2019	Breast cancer Malignant melanoma	Human	Melittin	Synergic effect with cold atmospheric plasma	Cytotoxic
30	Lim [56]	2019	Malignant melanoma	Human	BV, melittin	Inhibition of PI3K/AKT/mTOR and MAPK pathways. Upregulation of Cas3, Cas9.	Apoptosis, inhibition of migration and invasion
31	Shiassi Arani [57]	2019	Breast cancer	Mouse	BV	Unknown	Cytotoxic, synergy with cisplatin
32	Jung [58]	2018	TNBC	Human	BV	Reduction of Cas8, Cas9, Cas3 and PARP expression. Effect of cell morphology, DNA and protein fragmentation.	Apoptosis
33	Zarrinnahad [59]	2018	Cervical cancer	Human	Melittin	Unknown	Apoptosis
34	Khamis [60]	2018	Breast cancer	Human	BV	Upregulation of Bax, downregulation of Bcl2, EGFR, ERα. Influence of cell cycle.	Cytotoxic, synergy with hesperidin and piperine
35	Mohseni-Kouchesfahani [61]	2017	Acute myeloid leukemia	Human	BV	Unknown	Cytotoxic
36	Zhang [62]	2017	Non-small cell lung cancer	Human	Melittin	Decreasing of HIF-1α and VEGF level	Apoptosis, migration inhibiting
37	Alonezi [63]	2017	Ovarian cancer	Human	Melittin	Reduction in the levels of metabolites in TCA cycle, oxidative phosphorylation, purine and pyrimidine metabolism, and the arginine/proline pathway	Cytotoxic, synergy with cisplatin
38	Wang [64]	2017	Breast cancer	Human	Melittin	Downregulating CD147 and MMP-9 expression	Inhibition of migration and invasion
39	Alonezi [65]	2016	Ovarian cancer	Human	Melittin	Reduction in amino acids in the proline/glutamine/arginine pathway. Decreased levels of carnitines, polyamines, adenosine triphosphate (ATP) and nicotinamide adenine dinucleotide (NAD+).	Cytotoxic
40	Drigla [66]	2016	Breast cancer	Human	BV	Unknown	Antiproliferative
41	Gajski [67]	2016	Glioblastoma	Human	BV	Unknown	Cytotoxic, synergy with cisplatin
42	Zheng [68]	2015	Colon cancer	Human	BV	Increasing in DR4, DR5, p53, p21, Bax, cleaved caspase-3, cleaved caspase-8, and cleaved caspase-9 expression. NF-κB inhibition.	Apoptosis, growth inhibition
43	Mahmoodzadeh [69]	2015	Gastric cancer	Human	Melittin	Unknown	Necrosis
44	Kim [70]	2015	Cervical cancer	Human	BV	Inhibition of HPV E6 and E7 expression	Cytotoxic, antiviral
45	Choi [71]	2014	Non-small cell lung cancer	Human	BV	Increasing in DR3 expression. NF-κB pathway inhibition.	Apoptosis
46	Zhu [72]	2014	Esophageal squamous cell carcinoma	Human	Melittin	Influence of Tax and Bcl-2 proteins. Lack influence of cell cycle.	Apoptosis, radiosensitization of cells
47	Zhang [73]	2014	Liver cancer	Human	Melittin	CyclinD1 and CDK4 downregulation. Upregulation of PTEN. Attenuation of HDAC2 expression. PI3K/Akt signaling pathways inhibition.	Apoptosis
48	Jeong [74]	2014	Breast cancer	Human	Melittin	Inhibition of EGF-induced MMP-9 expression. Inhibition of NF-κB and PI3K/Akt/mTOR pathway. Inhibition mTOR/p70S6K/4E-BP1 pathway.	Apoptosis, inhibition of migration
49	Hoshina [75]	2014	Liver immortal cells	Human	BV	Unknown	Induction of genotoxicity and mutagenicity in human cells
50	Kollipara [76]	2014	Non-small cell lung cancer	Human	BV	Increasing in DR3, DR6, Fas, Bax, cleaved caspase-3, cleaved caspase-8	Enhancement of cytotoxicity against tumor of natural killer cells, apoptosis
51	Safaeinejad [77]	2014	Leukemia	Human	BV	Morphological changes, caspase-3-independent apoptosis	Potentiation of a novel palladium (II) complex, anti-proliferative, apoptosis
52	Shin [78]	2013	Cervical cancer	Human	Melittin	Decreasing of VEGF secretion, HIF-1ɑ inhibition	Inhibition of angiogenesis

**Table 2 cancers-15-03714-t002:** Original research on bee venom or melittin impact on cancer cells in animal models published in 2013–2023, present in the PubMed database. BV—bee venom, CAT—catalase, DR—death receptor, MDA—malonyldialdehide, MMP—matrix metalloproteinase, NF-κB—nuclear factor kappa B, NO—nitric oxide, SOD—superoxide dismuthase, TNF—tumor necrosis factor, VEGF—vascular endothelial growth factor.

No.	Author	Year	Cancer	Species	BV Component or Crude BV	Mechanism of Acting	Effect
1	Rocha [79]	2022	Colorectal cancer	Mouse	Melittin	Unknown	Inhibition of metastasis growth
2	El-Beltagy [80]	2021	Ovarian cancer Breast cancer	Rat	BV	Decreasing of serum MMP1, NF-κB, and TNF. Increasing in caspase 3. Influence on MDA, SOD, CAT.	Restoration of histological changes
3	El Bakary [81]	2020	Ehrlich ascites carcinoma	Mouse	BV, melittin	Decrease of Cas3, MMP2 and MMP9 activities. Decrease of TNF, VEGF, and NO levels.	Suppression of tumor proliferation, inhibition of angiogenesis
4	Lee [82]	2017	Lung carcinoma	Mouse	Melittin	Decrease the macrophage count in tumor environment, reduction of VEGF and CD206 expression in bone marrow-derived M2 macrophages	Reduction of tumor size, antiangiogenic effect
5	Zhang [62]	2017	Non-small cell lung cancer	Mouse	Melittin	Unknown	Inhibition of tumor growth
6	Lee [83]	2015	Cervical cancer	Mouse	BV	Increasing in FAS, DR3 and DR6 expression. Inhibition of NF-κB pathway.	Apoptosis

## Data Availability

The data are available upon request.

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
