# Peer review of "Can Bee Venom Be Used as Anticancer Agent in Modern Medicine?"

_cancers, 2023, doi:10.3390/cancers15143714_

Round 1
Reviewer 1 Report
It is an interesting review on bee venom (BV) and on the possibility of using it as an antitumor agent. It reviews the literature in which various therapeutic activities are described and also focus on the antitumor activity of BV.
However, there are some issues.
Major points
In order to suggest carry out a clinical trial, it is necessary to know previously the toxic concentrations of the compound in vitro, not only in tumor cells, but also in normal cells.
Questions that need to be answered:
Is the antitumor activity of BV at low or high concentrations?
Is the cytotoxicity of BV in normal non-tumor cells known?
Are the cytotoxic concentrations of BV in normal cells greater than, equal to, or less than those in tumor cells?
What would be the therapeutic doses (mg/kg body weight) of BV extrapolating the in vitro (micromolar) concentrations of BV?
Are the doses toxic to the body or are they within the therapeutic range?
All these points should be included and discussed in the text.
In the text there are only 78 references. However, in the references section there are 111. References 79-111 in the text do not appear. All references must be included in the text.
Minor points
- Line 18: what is the apitoxin? this should be clarified in text.
- Line 20: what does BV mean? this should be clarified in text.
Either the abbreviation is introduced the first time it appears in the abstract or the abbreviation is changed as in the rest of the abstract.
- Line 21: what is the mellitin? this should be clarified in the text.
- Line 106: severe clinical signs, it is not correct (the clinic has two components signs and symptoms). It should be changed for severe clinical effects.
- Line 149: Filety it's a typo. It should be changed by Filetype.
Author Response
Dear Reviewer,
Thank you for sending detailed and valuable reviews of our work. We hope that the corrections and additions have improved the quality of the manuscript. Below are answers to your questions. Corrections in the text are marked in red.
In order to suggest carry out a clinical trial, it is necessary to know previously the toxic concentrations of the compound in vitro, not only in tumor cells, but also in normal cells.
Answer: The text regarding the doses of BV exerted the cytotoxic effect on cancer cells has been added in the paragraph Can BV cure the cancer?. As noted, two research strategies are used in the literature; either the obtained effect is compared with the effect exerted on physiological cells (the study includes both tumor and physiological cell lines), or the comparison of the venom effect with the effect exerted by known cytostatic (the experiment does not use physiological cell lines).
Is the antitumor activity of BV at low or high concentrations?
Answer: The cytotoxic concentration of crude venom against cancer cells starts at 1 µg/ml medium (in vitro). Similar concentrations have been used in in vivo animal studies (0.5-1 mg/kg bw).
Is the cytotoxicity of BV in normal non-tumor cells known?
Answer: Yes, the cytotoxicity of BV is also against physiological cells, the same as known cytostatic drugs. However, in the comparison studies the effect of BV on cancer cells was clearer, increased more rapidly, and started from lower BV concentrations.
Are the cytotoxic concentrations of BV in normal cells greater than, equal to, or less than those in tumor cells?
Answer: The cytotoxic effect of BV depends on both the incubation time and the type of cell line. Not every study compared the effect of venom on tumor cells vs. physiological cells. However, in studies that included such comparisons, the cytotoxic effect of the venom was revealed at lower concentrations in relation to the tumor cell than in relation to physiological ones. Also, the increase in the effect was more pronounced against cancer cells.
What would be the therapeutic doses (mg/kg body weight) of BV extrapolating the in vitro (micromolar) concentrations of BV?
Answer: The cytotoxic effect of BV on tumor cells is revealed at a concentration of >1 µg/ml of the medium (depending on the type of cell line). Theoretically, 75 kg human needs approximately 75 mg of venom to reach similar concentration of the drug. In animal studies, a venom dose about 1.0 mg/kg bw was used which in the case of a 75 kg human gives a dose 75 mg of venom (comparable to the previous results). Therefore, the results are promising to consider starting clinical trials in the future.
Are the doses toxic to the body or are they within the therapeutic range?
Answer: It seems that the stings of even a few bees are not able to reach concentrations having a therapeutic effect in in vitro studies. However, due to promising results in studies on cell lines, future research is aimed at reducing the side effects of BV therapy in order to safely increase its concentration in the body.
All these points should be included and discussed in the text.
In the text there are only 78 references. However, in the references section there are 111. References 79-111 in the text do not appear. All references must be included in the text.
Answer: References 85-113 appear only in Table 1 or Table 2. Unfortunately, the tables were attached to the submission in separate files and were not included into the main text. In the current version, the tables are already in the text.
Minor points
- Line 18: what is the apitoxin? this should be clarified in text. - Line 20: what does BV mean? This should be clarified in text.
Answer: Apitoxin and BV abbreviation is clarified in the sentence: Natural bee products used in the treatment and prevention of various diseases include: honey, bee pollen, propolis, royal jelly, bee pollen, beeswax, as well as bee venom (BV) also known as apitoxin [2].
Either the abbreviation is introduced the first time it appears in the abstract or the abbreviation is changed as in the rest of the abstract.
Answer: The abstract was corrected.
- Line 21: what is the mellitin? this should be clarified in the text.
Answer: The mellitin was corrected to melittin.
- Line 106: severe clinical signs, it is not correct (the clinic has two components’ signs and symptoms). It should be changed for severe clinical effects.
Answer: The text was corrected.
- Line 149: Filety it’s a typo. It should be changed by Filetype.
Answer: The word was corrected.
Reviewer 2 Report
The objective of the authors was to provide a comprehensive review on bee venom (BV) properties with particular emphasis on BV therapeutic implications in the treatment of cancer. The authors highlighted the importance of natural products in medicine and emphasized the chemical biodiversity and therapeutic potential of bee products, particularly BV. The authors have also outlined the chemical composition of BV, including its major components melittin and phospholipase A2. In addition the authors have highlighted the cytotoxic effects of BV and melittin on cancer cells and their ability to induce apoptosis, inhibit proliferation, and suppress cell migration. Furthermore, the authors have also discussed the impact of BV on specific signaling pathways implicated in cancer such as PI3K/Akt/mTOR and MAPK pathways. It is interesting article but the manuscript could not be considered for publication in the current form for following reasons.
Major concerns:
1) Could authors please elaborate on the safety profile of using bee venom as a therapeutic agent. The authors briefly mentioned the potential side effects and allergic reactions associated with bee venom and it would be beneficial if authors could elaborate on potential risks in clinical settings and contraindications.
2) Could authors also discuss about limitations, challenges/ potential drawbacks and future directions.
Minor concerns:
1) Please correct the typo "Filety" to "Fifty" on page # 5, Ln # 149
Please proof read the manuscript for syntax errors.
Author Response
Dear Reviewer,
Thank you for sending detailed and valuable reviews of our work. We hope that the corrections and additions have improved the quality of the manuscript. Below are answers to your questions. Corrections in the text are marked in red.
1) Could authors please elaborate on the safety profile of using bee venom as a therapeutic agent. The authors briefly mentioned the potential side effects and allergic reactions associated with bee venom and it would be beneficial if authors could elaborate on potential risks in clinical settings and contraindications.
Response: The most dangerous complication of contact with bee venom is anaphylactic shock. However, according to literature, no parameter has been identified that may predict which sensitized people will have a future systemic sting reaction. The only way to reduce the body’s response to BV is to immunize in advance. Observations with beekeepers show that frequent bee stings reduce serious adverse symptoms. Perhaps patients will require pre-immunization before starting therapy with therapeutic doses of bee venom. To our knowledge, clinical trials are not yet conducted on this topic.
2) Could authors also discuss about limitations, challenges/ potential drawbacks and future directions.
Response: Venom is a toxic substance with the ability to elicit various responses from the body. However, promising results obtained in in vitro and preclinical studies contribute to the search for a safe way to deliver BV to the tumor. Future directions involve either reducing the side effects of the venom by loading it onto nanoparticles or trying to deliver the venom directly to the tumor. Both methods of treatment have been discussed in the literature, which is why we only mentioned such directions in our work.
Minor concerns:
1) Please correct the typo “Filety” to “Fifty” on page # 5, Ln # 149
Response: The error was corrected.
Comments on the Quality of English Language
Please proof read the manuscript for syntax errors.
Response: The text was corrected for syntax errors.
Round 2
Reviewer 1 Report
Now everything is OK.